# Structure of *Escherichia coli* cytochrome *bd*-II type oxidase with bound aurachin D

Antonia Grauel[1,7], Jan Kägi[1,7], Tim Rasmussen [2], Iryna Makarchuk[3], Sabrina Oppermann[1], Aurélien F. A. Moumbock [4], Daniel Wohlwend [1], Rolf Müller [5,6], Frederic Melin [3], Stefan Günther [4], Petra Hellwig [3], Bettina Böttcher [2✉] & Thorsten Friedrich [1✉]

Cytochrome *bd* quinol:$O_2$ oxidoreductases are respiratory terminal oxidases so far only identified in prokaryotes, including several pathogenic bacteria. *Escherichia coli* contains two *bd* oxidases of which only the *bd*-I type is structurally characterized. Here, we report the structure of the *Escherichia coli* cytochrome *bd*-II type oxidase with the bound inhibitor aurachin D as obtained by electron cryo-microscopy at 3 Å resolution. The oxidase consists of subunits AppB, C and X that show an architecture similar to that of *bd*-I. The three heme cofactors are found in AppC, while AppB is stabilized by a structural ubiquinone-8 at the homologous positions. A fourth subunit present in *bd*-I is lacking in *bd*-II. Accordingly, heme $b_{595}$ is exposed to the membrane but heme *d* embedded within the protein and showing an unexpectedly high redox potential is the catalytically active centre. The structure of the Q-loop is fully resolved, revealing the specific aurachin binding.

[1] Institut für Biochemie, Albert-Ludwigs-Universität Freiburg, Freiburg, Germany. [2] Biocenter and Rudolf Virchow Center, Julius-Maximilians-Universität Würzburg, Würzburg, Germany. [3] Laboratoire de Bioélectrochimie et Spectroscopie, UMR 7140 CMC, Université de Strasbourg, CNRS, Strasbourg, France. [4] Institut für Pharmazeutische Wissenschaften,  Albert-Ludwigs-Universität Freiburg, Freiburg, Germany. [5] Department of Microbial Natural Products, Helmholtz Institute for Pharmaceutical Research Saarland, Saarbrücken, Germany. [6] Helmholtz Centre for Infection Research and Department of Pharmacy at Saarland University, Saarbrücken, Germany. [7] These authors contributed equally: Antonia Grauel, Jan Kägi. ✉email: Bettina.Boettcher@uni-wuerzburg.de; Friedrich@bio.chemie.uni-freiburg.de

Cytochrome *bd* quinol:dioxygen oxidases (called *bd* oxidases for simplicity hereafter) are terminal reductases exclusively found in bacterial and archaeal respiratory chains. They couple quinol oxidation and release of protons to the periplasmic side with proton uptake from the cytoplasmic side to reduce dioxygen to water[1–3]. Thus, *bd* oxidases contribute to the generation of a protonmotive force (pmf) by a vectorial charge transfer[4,5]. They display a high affinity towards dioxygen, enabling growth under microaerobic conditions and endow pathogens such as *Shigella flexneri*[6], *Mycobacterium tuberculosis*[7], and various streptococcus species[8] with resistance to intracellular stressors such as NO and intracellular hypoxia, making them excellent drug targets. Furthermore, *bd* oxidases play a key role in tolerance to or degradation of NO[9], $H_2S$[10], CO[11] and other gaseous ligands that are involved in anti-inflammatory immune response.

The *Escherichia coli* aerobic respiratory chain contains three terminal oxidases, cytochrome $bo_3$ oxidase, cytochrome *bd*-I oxidase (*bd*-I) and cytochrome *bd*-II oxidase (*bd*-II)[2,12,13]. While the cytochrome $bo_3$ oxidase belongs to the unrelated family of heme-copper oxidases, the *bd* oxidases display distinct features such as the presence of a *d*-type heme in the active site and the lack of copper atoms. Expression of *cyoABCDE*, encoding $bo_3$ oxidase is activated at full aerobic conditions, while *cydABX*, coding for *bd*-I, is maximally expressed at microaerobic and oxygen-rich conditions[14]. In contrast, *bd*-II, encoded by the *appCBX* operon, is expressed under complete anaerobiosis[14] and upon entry into the stationary phase and phosphate starvation[14–17].

While the *E. coli bd*-II has been discovered quite recently and is relatively poorly characterized[18], the *bd*-I is well-studied. It is made up of the two large subunits CydA and CydB, each comprising nine transmembrane (TM) helices, and the single TM helix subunit CydX[19–22]. CydA harbors all heme groups of *bd*-I, namely the low-spin heme $b_{558}$, acting as electron input device to catalyze quinol oxidation[23,24], the high-spin heme $b_{595}$ expected to deliver electrons to the third, the unique *d*-type heme, which is the site where dioxygen is reduced to water. CydB is related to CydA by a pseudo twofold symmetry and contains a structural ubiquinone-8 at a position occupied by the hemes in CydA[21,22]. CydA contains a soluble, periplasmic domain of variable length, termed Q-loop. This domain is expected to be involved in quinol binding and oxidation[21–24]. In *E. coli*, CydX plays a structural role as it is essential for the stability and/or the assembly of the active site[20]. Unexpectedly, *E. coli bd*-I contains a fourth subunit, CydY or CydH, that is encoded by the orphan gene *ynhF*[21,22]. CydY (or CydH) is also a single TM helix protein and probably plays a functional role by blocking substrate access to heme $b_{595}$ from the membrane[21,22]. The structure of *bd*-I from *Geobacillus thermodenitrificans* revealed that this enzyme lacks the fourth subunit and, accordingly, the positions of hemes $b_{595}$ and *d* are interchanged while the overall architecture of the enzyme remained unaltered[25]. To gain access to the substrate dioxygen, *E. coli bd*-I contains a hydrophobic channel, leading from CydB to heme *d* on CydA[21,22].

By contrast, much less is known about *bd*-II. It is encoded by the *appCBX* genes, leading to the production of subunits AppC (predicted molecular mass: 58.1 kDa), AppB (42.4 kDa) and AppX (3.6 kDa)[18]. It also contains heme $b_{558}$, $b_{595}$ and *d* as cofactors[26]. The two large subunits of *bd*-I and *bd*-II share high sequence similarity of about 60%[5]. The *appBCX* genes are most likely crucial for gut colonization by *E. coli* and other inflammatory Enterobacteriaceae family members[27].

Here, we present the 3.0 Å resolution structure of *E. coli bd*-II with bound aurachin D, a specific inhibitor of quinol oxidation by *bd* oxidases, as obtained by electron cryo-microscopy (cryo-EM).

The structure comprises three subunits, AppB, AppC and AppX that show a similar architecture as the homologous subunits in *bd*-I. The arrangement of the heme cofactors in AppC corresponds to that of the hemes in *bd*-I. However, due to the lack of a fourth subunit, heme $b_{595}$ is accessible from the membrane but it is not the active site. A narrow substrate channel leads from AppB to heme *d* on AppC. The binding site of aurachin D is clearly resolved and aurachin binding enables structure determination of the entire Q-loop.

## Results

### Gene expression and protein production, purification and characterization.
*E. coli* strain BL21*Δ*cyo*[28] was transformed with pET28b(+) *appC*$_{his}$*BX* encoding *bd*-II and grown under oxic conditions. Cells were disrupted and cytoplasmic membranes were prepared by differential centrifugation. Membrane proteins were extracted with the detergent lauryl maltose neopentyl glycol (LMNG) and purified by affinity- and size exclusion-chromatography (Supplementary Fig. 1). The final chromatogram showed two peaks, with the second one containing *bd*-II (Supplementary Fig. 1). The major bands of the gel were identified as AppC and AppB by mass spectrometry (AppC: sequence coverage: 33%, overall score 69; AppB: sequence coverage: 3%, overall score 19). The identity of subunit AppX, migrating at an apparent molecular mass of about 4 kDa on a 16% SDS-gel (Supplementary Fig. 1), was also established by mass spectrometry (sequence coverage: 20%, overall score 25). Importantly, no other protein with a molecular mass below 10 kDa was detected in the low molecular mass region of the 16% SDS-gel by mass spectrometry.

From 9.6 L medium, 40 g cells and approximately 2 mg *bd*-II were obtained. The (reduced-minus-oxidized) difference spectrum of the preparation showed the typical absorbance of hemes $b_{558}$, $b_{595}$ and *d* in a 1:1:1 stoichiometry using the given extinction coefficients[2] (Fig. 1 and Supplementary Fig. 1). The single γ-peak in the Soret region at 440 nm matched the properties of hemes $b_{558}$ and $b_{595}$. In addition, heme $b_{558}$ gave rise to absorbance at 533 and 563 nm, while heme $b_{595}$ produced another absorbance peak at 595 nm. The signal at 630 nm was finally assigned to heme *d*. The negative absorbance at 657 nm derived from the ferrous oxygenated heme *d*. Thus, the spectral properties of *bd*-II match those of *bd*-I (Fig. 1).

The UV-vis absorbance of the cofactors was used to determine their redox potentials by an electrochemical titration in a thin-layer cell (Fig. 1 and Supplementary Fig. 2)[29]. From their absorbance at the α-band at 563 nm, the redox potential of both *b*-type hemes was determined to +237 mV (Fig. 1). It was not possible to further distinguish their individual redox potentials. This indicates that they exhibit similar values differing by less than 100 mV. The redox potential of heme *d* was determined from its absorbance at 629 nm to +440 mV. The hysteresis between oxidative and reductive titrations was <40 mV indicating the lack of cooperativity or other coupled processes (Supplementary Fig. 2). Taken together, the redox potentials of the *bd*-II cofactors are 60–80 mV (*b*-type hemes) and 180 mV (heme *d*) more positive than those of the corresponding cofactors in *bd*-I (Fig. 1)[30].

### Structure of *E. coli bd*-II oxidase.
According to its elution volume and BN-PAGE analyses, *bd*-II in LMNG eluted as a monomer from size-exclusion chromatography (Supplementary Fig. 1b). The protein was concentrated to 24 mg/mL, shock frozen in liquid nitrogen and stored at −80 °C. Aliquots for cryo-EM preparation were thawed and diluted to 5.6 mg/mL in course of the amphipol exchange. Prior to cryo-EM analysis, purified *bd*-II

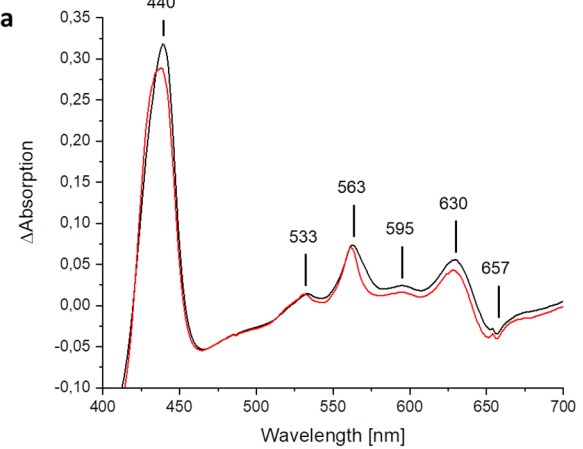

**b**

| | Redoxpotential [mV] | | |
|---|---|---|---|
| | heme $b_{558}$ | heme $b_{595}$ | heme $d$ |
| *bd*-II | +237 | +237 | +440 |
| *bd*-I | +168 | +176 | +258 |
| *G. th* | +50 | +155 | +15 |
| *C. glu* | +102 | +160 | +228 |

**Fig. 1 Spectral and thermodynamic properties of the heme groups. a** UV-vis (reduced-minus-oxidized) difference spectra of the preparation of *bd*-II (black) and *bd*-I (red). The absorbance of the individual heme groups is indicated. **b** Redox potential of the heme groups of various *bd* oxidases. The values for the oxidases other than *bd*-II were taken from ref. [30].

was incubated with 160 μM aurachin D, a threefold molar excess of the specific inhibitor of *bd* oxidases (see below). On the cryo-grid, predominantly dimeric species of *bd*-II were detected. Peaks of *bd*-II monomers and dimers were also observed by mass photometry when placing 1 μL of a concentrated sample (24 mg/mL) on a 20 μL drop of the buffer used for grid preparation on the cover slip (Supplementary Fig. 3)[31]. Although unlikely[31], we cannot completely exclude the possibility that the presence of amphipol led to protein dimerization. The final cryo-EM map provided sufficient details for modeling the structure of the *bd*-II dimer with the bound inhibitor aurachin D at a resolution of 3.0 Å (Fig. 2, Supplementary Table 1 and Supplementary Figs. 4 and 5). The homo-dimer contains two oxidase hetero-trimers related by twofold symmetry (Fig. 2). Each hetero-trimer consists of the three subunits AppB, AppC and AppX of which the small AppX form the major dimer contact between the two hetero-trimers. AppC and AppX establish additional contacts between the monomers through hydrophobic interactions between Val41$^{AppC}$ and Trp23$^{AppX}$, as well as between Val42$^{AppC}$ and Val22$^{AppX}$ and Trp23$^{AppX}$, respectively.

Within each hetero-trimer, the large subunits AppB and AppC are related by a pseudo-twofold symmetry axis similar as CydA to CydB from *bd*-I. AppB and AppC share the same fold with nine TM helices that are arranged in two four-helix bundles and a peripheral helix. The interaction between AppB and AppC is mediated via the pseudo-symmetry related hydrophobic residues in TM helices 2, 3 and 9. Based on the structural conservation between AppB and AppC, it is reasonable to assume that *appB* and *appC* genes originated from gene duplication, although this is not reflected in their primary sequence (Supplementary Fig. 6).

The single helix subunit AppX has a length of 30 amino acid residues and is completely buried within the membrane. It interacts with the AppC TM helices 1, 5 and 6, most likely

stabilizing the protein fold to bind hemes $b_{595}$ and *d*. The dimerization of *bd*-II by AppX is mediated by several weak hydrophobic interactions, involving Met1, Leu4, Leu5, Val8, Leu11, Leu12, Ser15, Leu16, and Leu19. The Leu residues of the helix connect both monomers in a 'leucine zipper-type' fashion. Leu4 and Leu16 are both replaced by Phe residues in the homolog CydX in *bd*-I, while Leu5 and Leu19 are exchanged to Ala and Ile residues, respectively (Supplementary Fig. 6). These changes might prevent dimer formation in *bd*-I. Indeed, the positions of all Leu residues are conserved in the homologs of AppX identified by their organization in a *bd*-II operon. In contrast, only the five Leu residues found in CydX are conserved among the CydX homologs, partly in different positions compared to AppX (Supplementary Fig. 6). However, whether *bd*-II is also a dimer in the *E. coli* cytoplasmic membrane remains an open question.

AppB features a long hydrophobic patch in a cleft towards the periplasmic space filled with an extended electron density that was perfectly fitted as ubiquinone-8 (Fig. 2c). Its position follows the arrangement of the heme groups in AppC. The quinoid headgroup points away from AppC, indicating a structural role of bound quinone in stabilizing the AppB fold as reported for CydB[21,22].

**The Q-loop and binding of aurachin D**. The family of *bd* oxidases is divided into L (long) and S (small)-subfamilies depending on the length of the periplasmic Q-loop[2]. Members of the L-subfamily harbor a C-terminal insertion in the Q-loop of about 60 residues that is essential for structural stability[32,33]. The N-terminal part of the Q-loop is involved in quinone binding[21]. The Q-loop of AppC (Fig. 3a, b) is located between TM helices 6 and 7 and extends over a length of 134 residues including the linker to the residual protein (Fig. 3a, b and Supplementary Fig. 7), rendering *E. coli* *bd*-II a member of the L-subfamily. Here, we resolve the entire Q-loop of *bd*-II covering positions 262−302 of the C-terminal region that are not found in the structures of the other *bd* oxidases[21,22,25]. The Q-loop consists of six small helices (Q$_{α1}$, P250-E257; Q$_{α2}$, L292-T297; Q$_{α3}$, L307-S332; Q$_{α4}$, P338-R345; Q$_{α5}$, L351-Y360, and Q$_{α6}$, A369-A379) that are connected by short loops (Fig. 3a, b). An additional density was identified in the cavity of the Q-loop that was modeled as a bound aurachin D molecule.

Aurachins inhibit electron transfer from ubiquinol-8 to heme $b_{558}$ by blocking the quinol binding site that is located close to the membrane surface (Fig. 3a, b)[34–37]. Although aurachin C and D bind with high affinity to *bd* oxidases, a reliable IC$_{50}$ value for *bd*-II has not yet been determined[38]. Hence, we titrated the duroquinol:dioxygen oxidoreductase activity of our *bd*-II preparation with increasing amounts of aurachin C and D and determined the apparent IC$_{50}$ to 7.1 and 11.1 nM, respectively (Supplementary Fig. 8). Due to the extremely high affinity of aurachin D, the preparation of *bd*-II was incubated with only a threefold molar excess of aurachin D just prior to blotting on the cryo-EM grid.

The strong binding of aurachin is mostly due to a structural complementarity to the surface of the quinol binding site (Fig. 3c, d). Interactions with Phe269, Val271, Leu291, and Leu295 (the latter two from Q$_{α2}$) stabilize the Q-loop and it is apparently this interaction with aurachin D that allowed for full resolution of the entire Q-loop (Fig. 3e). Aurachin D is bound by seven additional hydrophobic interactions to AppC (Fig. 3e). One H-bond of the aurachin nitrogen atom to Asp239 of AppC adds to the strong binding (Fig. 3e). This residue is conserved among *bd*-oxidases (Supplementary Fig. 6). To experimentally test its importance for quinol oxidation and aurachin binding we produced the D239N$^{AppC}$ variant in strain CBO and purified wild-type *bd*-II

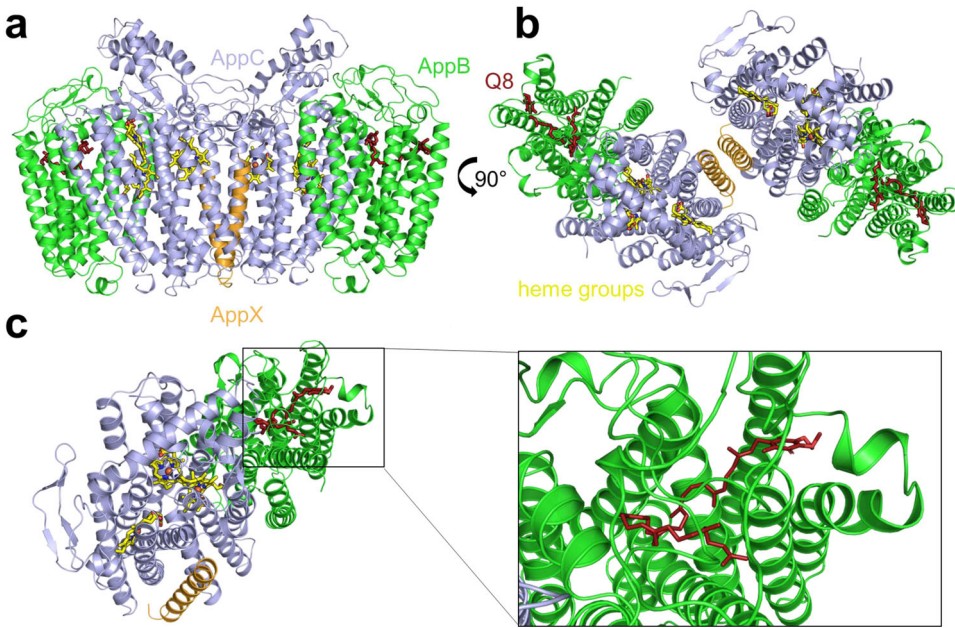

**Fig. 2 Cryo-EM structure of the *E. coli bd*-II dimer at 3.0 Å resolution. a** Front view with AppC (blue) and AppB (green), comprising nine TM helices, each. AppX (orange) is a single TM helix that stabilizes the four-helix bundles, coordinating hemes $b_{595}$ and $d$ (yellow). The positions of heme $b_{558}$ (yellow) and of ubiquinone-8 (red) are indicated. **b** Top view on the dimer with fully resolved Q-loop. **c** Position of bound ubiquinone-8 (red, shown as sticks). The inset shows how the quinone is bound to 6 helices of subunit AppB.

and the D239N$^{AppC}$ variant from that strain by the procedure described above. Due to the presence of the chromosomal *appCBX* operon, the variant could not be produced in strain BL21*Δ*cyo*. The proteins produced in strain CBO eluted at identical positions from the chromatographic columns. Remarkably, the enzymatic duroquinol oxidase activity of the D239N$^{AppC}$ variant was only 23% of that of the wild-type protein (2.91 vs. 12.84 μmol/mg/min). Furthermore, the IC$_{50}$ decreased from 11.1 to 60.7 nM (Supplementary Fig. 8). Thus, Asp239 on AppC is strongly involved in aurachin binding and quinol oxidation. Binding of aurachin D completely blocks access to heme $b_{558}$, the primary electron acceptor from the quinol (Fig. 3c, d) and, thus, prevents further electron transfer to hemes $b_{595}$ and $d$.

**Arrangement of heme groups and access to the active site**. The heme groups $b_{558}$, $b_{595}$, and $d$ are positioned in between the TM helices of AppC and form a triangular arrangement (Fig. 4a). Heme $b_{558}$ is coordinated by His186 and Met393 as axial ligands, heme $b_{595}$ by Glu445 and heme $d$ by His19. The iron−iron distance between heme $b_{558}$ and heme $d$ is 18.8 Å, that between hemes $b_{558}$ and $b_{595}$ is 15.6 Å and the one between heme $b_{595}$ and $d$ amounts to 11.1 Å (Fig. 4a). The position of Trp441, being discussed as important for intramolecular electron transfer[39] and formation of the F$^{+•}$ intermediate, is conserved in comparison with the *bd*-I type oxidase. Likewise, this arrangement of the heme groups in *bd*-II is virtually identical to that of the heme positions within *E. coli bd*-I. Noteworthy, hemes $b_{558}$ and $b_{595}$ are accessible from the membrane. For heme $b_{558}$ this is not surprising as it is the primary electron acceptor from the substrate quinol[21–25]. However, while in *E. coli bd*-I heme $b_{595}$ is shielded from the membrane by the additional fourth subunit CydY (or CydH), a fourth subunit was not detectable by mass spectrometry of the *bd*-II preparation in solution and of the low molecular weight region of a 16% SDS-gel (Supplementary Fig. 1). It has been reported that *G. thermodenitrificans bd* oxidase also lacks a

homolog of CydY[25], but in comparison with *E. coli bd*-II, the positions of hemes $b_{595}$ and $d$ are interchanged there. Consequently, heme $b_{595}$ of *E. coli bd*-II is found in a position that corresponds to the active site in the *G. thermodenitrificans* enzyme, rendering the membrane accessibility plausible. However, the architecture around *E. coli bd*-II heme $d$ implies that this is the active site, because the TM helix 3 of AppC exerts a strong curvature (Figs. 2 and 4). This curvature is due to the insertion of the additional Leu101 also found in *E. coli bd*-I but lacking in *G. thermodenitrificans bd* oxidase (Supplementary Fig. 6). This insertion places Glu99 in 5.1 Å coordination distance to the central Fe of heme $d$, thus providing space for substrate binding.

To experimentally address the question, whether heme $d$ or $b_{595}$ is the active site, the preparation was incubated with KCN and UV-vis difference spectra were recorded over time (Fig. 4d). It is known that cyanide reacts with air-oxidized *bd*-I oxidase, when heme $d$ is accessible[40,41]. The reaction results in a displacement of the oxy-complex of heme $d$ by a cyano-complex that is not reduced by a mild excess of reductant[40,41]. The preparation of *bd*-II incubated with 0.5 mM KCN clearly showed the development of the heme $d$ cyano-complex at 652 nm (Fig. 4d). Furthermore, in agreement with the literature[40,41], it was not possible to reduce heme $d$ in the cyano-complex with a slight molar excess of dithionite (Fig. 4d). These data establish that heme $d$ is indeed the active site of *bd*-II. This finds further support in an extra electron density located in the putative substrate binding cavity between Glu74 and Glu94 in AppC that may be caused by a dioxygen or a water molecule (Fig. 4c).

Dioxygen has no access to heme $d$ via the solvent accessible channel to heme $b_{595}$ because heme $d$ itself blocks a deeper penetration of substrates to the active site. In *E. coli bd*-I, another long substrate channel for oxygen on the opposing side was described, leading from CydB to heme $d$ on CydA[21,22]. A similar channel is also present in *bd*-II, leading from AppB to Glu99 on AppC (Fig. 4b). However, this channel seems to have a smaller diameter, in particular due to the methyl group of Ala100 on AppC that narrows the channel diameter approximately by one

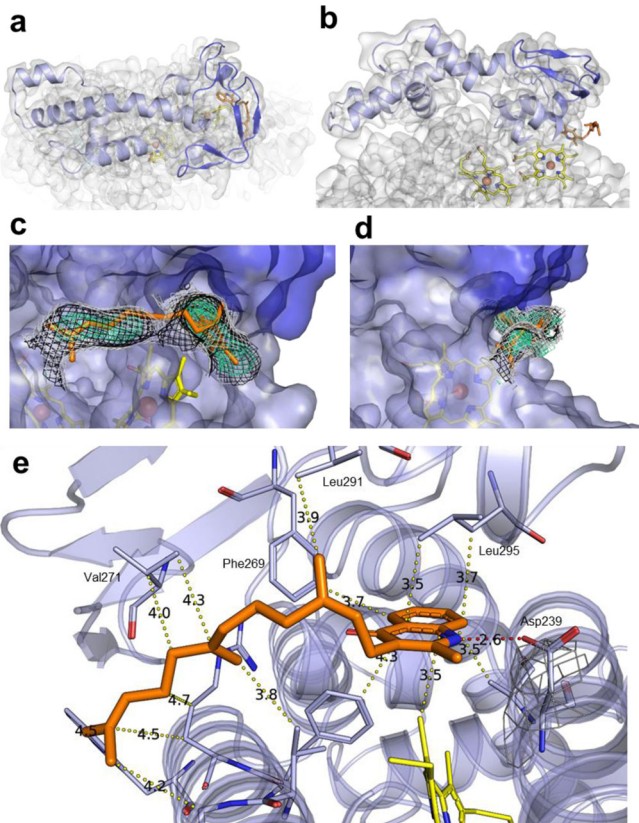

**Fig. 3 Structure of the Q-loop and binding of aurachin D. a** Top view and **b** side view of the entire Q-loop (blue) of *bd*-II on top of subunit AppC (gray) with the previously unresolved region in dark blue. The additional density, interpreted as aurachin D is shown in orange. **c, d** Surface view on the quinol binding site with the Q-loop in blue, the TM part of AppC in light blue, heme $b_{558}$ in yellow and bound aurachin D in orange. The experimental Coulomb density map is shown as different meshes in green cyan (at 1.0$\sigma$), black (at 0.65$\sigma$), and white (at 0.5$\sigma$), all carved at 2 Å radius around model atom centers. The view on the binding site is turned by 90° from (**c**) to (**d**). **e** Hydrophobic interactions and the H-bond of aurachin D with *bd*-II are indicated by dotted lines and are labeled with distances.

fourth. At the homologous position, the *bd*-I features a Gly residue that widens the channel. To further analyze the geometry of the putative oxygen channels of both oxidases, we probed the radii with the program Caver 3.0 [42]. The analysis of the derived models revealed radii of 1.4 Å for *bd*-I and of 1.2 Å for *bd*-II. Hence, despite the *bd*-II channel diameter being smaller, it is still sufficient to provide oxygen access to heme *d*.

This implies that Ala100 does not block the oxygen channel. However, if it would do so, the mutation of the homologous Gly100 in *bd*-I to Ala100 would significantly decrease, if not fully abolish the mutant's activity. Accordingly, we introduced the corresponding mutation (G100A$^{bd\text{-I}}$) into *cydA* of *bd*-I oxidase. The *bd*-I oxidase variant was prepared from the mutant strain as described[22]. The duroquinol:dioxygen oxidoreductase activity of the preparation of G100A$^{bd\text{-I}}$ variant ($11.7 \pm 0.3$ µmol/min/mg, standard deviation) was even slightly enhanced compared to that of *bd*-I ($8.6 \pm 2.2$ µmol/min/mg, standard deviation). Thus, the methyl group of Ala100 does not hamper access of oxygen to heme *d*. The duroquinol:dioxygen oxidoreductase activity of the *bd*-II preparation was determined to $3.3 \pm 0.2$ µmol/min/mg (standard deviation), which is roughly 2.5-fold slower than that of the *bd*-I oxidase in agreement with the higher redox potential of the heme groups (Fig. 1).

**Comparison to other *bd* oxidases**. The structure of the *E. coli bd*-II monomer is very similar to that of *E. coli bd*-I (Fig. 5a), not surprising, considering the high sequence similarity of about 60%[2] (Supplementary Fig. 6). The positions of AppX and CydX completely overlap, emphasizing their structural role in stabilizing the binding of heme *d* and $b_{595}$ (Fig. 6a). The additional role of AppX in mediating the interactions between the two *bd*-II monomers does not lead to a change of its structure, indicating that indeed the individual residues of the TM helix contribute to dimer formation. Accordingly, the three common subunits of *bd*-I and *bd*-II align with a root-mean-squared deviation (RMSD) of 0.73 Å for the $C_\alpha$ atoms. The preparation of *bd*-II lacks a homolog of the fourth subunit CydY that blocks the access to heme $b_{595}$ in *bd*-I. However, the relative tilt between TM helices 1 and 8 of AppC providing access to heme $b_{595}$ is lower than in *bd*-I and the channel to the heme is consequently more narrow. Still, it is accessible from the membrane as discussed above. Furthermore, the heme groups bind at very similar positions within the oxidases (Fig. 6b) as already indicated by the akin UV-vis redox difference spectra (Fig. 1). The same holds true for the position of the structural ubiquinone-8 that replaces the heme groups in AppB/CydB (Fig. 6c, d).

The structure of *E. coli bd*-II is also very similar to that of *G. thermodenitrificans bd* oxidase (Fig. 5b). Importantly, the access from the membrane to *E. coli* $b_{595}$ and *G. thermodenitrificans* heme *d* that are found in homologous positions differs from each other. According to the position of the protein backbone of the derived model, the access to the active site heme *d* in *G. thermodenitrificans* is provided by an open tunnel with a diameter of 6.6 Å (Supplementary Fig. 9). The homologous position in *bd*-II has a narrow keyhole shape with a maximum diameter of 5.5 Å (Supplementary Fig. 9). However, even this obstruction of the entry site is not sufficient to prevent a reaction of a substrate with heme $b_{595}$ in *bd*-II.

**Proton pathway**. The reaction of *bd* oxidases depends on a proton pathway that enables uptake of protons from the cytoplasmic site to heme *d* where dioxygen is reduced to water. TM helices 2 and 3 of AppC and of AppB, respectively, provide a broad cavity at the cytoplasmic site of the membrane (Fig. 7). Due to its hydrophilic nature this cavity is most likely filled with water molecules. A hydrophilic channel provided by the four helices protrudes from the large, hydrophilic cavity and extends nearly over the entire distance to heme *d*. Protons may also access heme *d* through Asp105 and Glu58 of AppC. From here, they can be transferred to the propionate group of heme *d*. The proton entry site of *bd*-II is much more open than those of the other oxidases (Fig. 7). The distance from Asp105 to the heme propionate group covers a distance of 13.1 Å. Interestingly, the channel to heme *d* is narrower in *bd*-I and *bd* from *G. thermodenitrificans* preventing access of water molecules as obvious from a Caver analysis[42]. Accordingly, more titratable side chains are needed to bring protons to heme *d*. The pathway for protons that leads along a series of titratable residues from Asp119 (*bd*-I) to the heme propionate covers a distance of 24.2 Å, while the corresponding one starting at His126 in *G. thermodenitrificans* has a length of even 30.1 Å.

**Discussion**

The structure of *E. coli bd*-II reported here shows an overall high similarity to that of *E. coli bd*-I[21,22]. However, there are few remarkable differences. First, *bd*-II consists of only three different subunits resulting in heme $b_{595}$ being accessible from the membrane (Supplementary Fig. 9). However, heme *d* is the active site, most likely caused by its unusually high redox potential (Figs. 1b

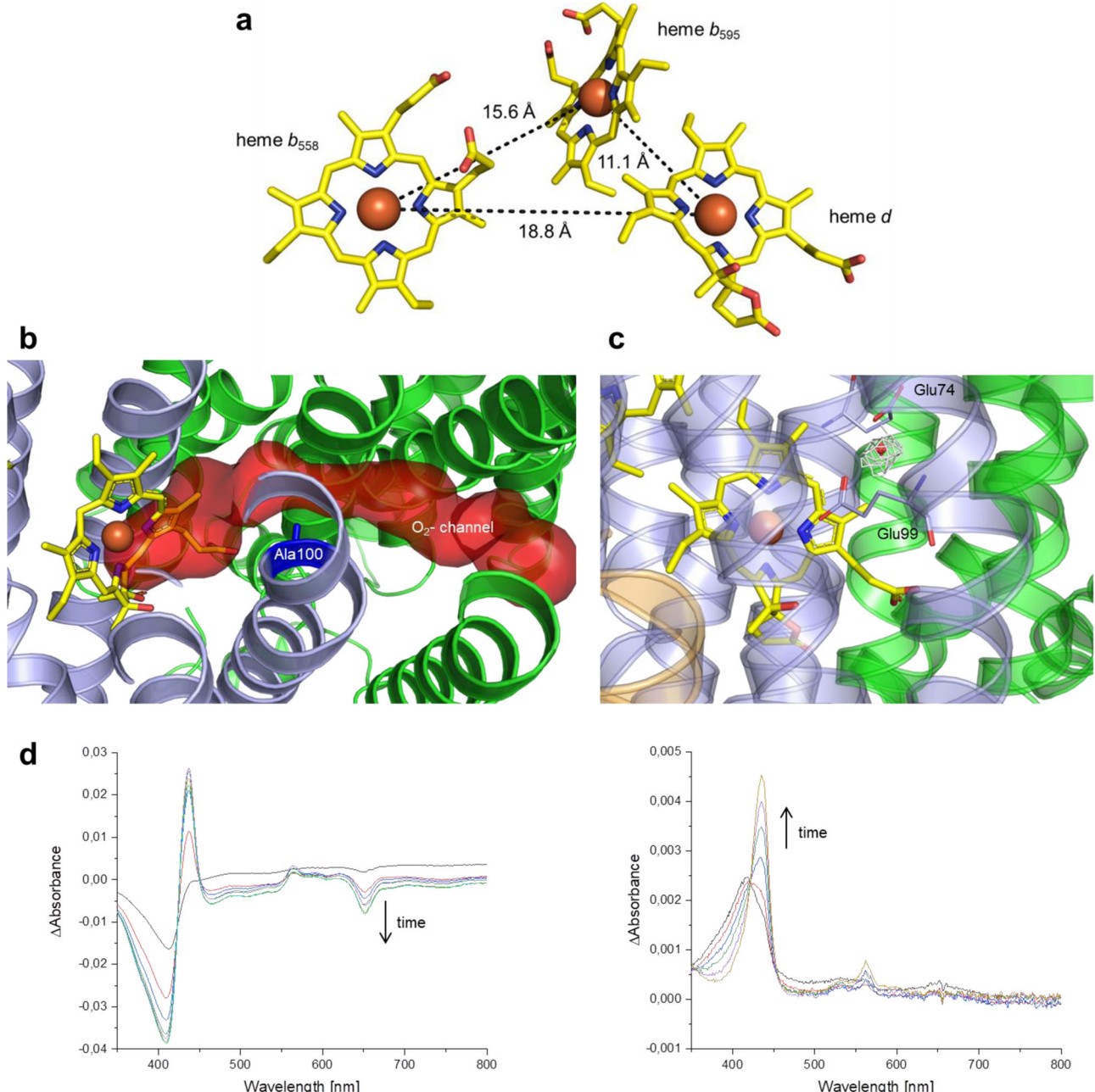

**Fig. 4 Arrangement of heme groups in *bd*-II and substrate binding. a** Arrangement of heme groups in AppC of *E. coli bd*-II. **b** Channel from AppB to heme *d*. The channel identified by Caver is shown in red. Ala100 (dark blue, shown as sticks) constricts the channel diameter. **c** Electron density next to heme *d* that could be caused by the substrate dioxygen. **d** UV-vis difference spectra of *bd*-II incubated with CN⁻ over 30 min (left) and reduction of the CN⁻-treated sample with dithionite (right).

and 4). This could indicate that both enzymes use different substrates under physiological conditions in accordance with the fact that the corresponding genes are differently expressed under different growth conditions[14–17]. Recently, the structure of the *Mycobacterium smegmatis bd*-oxidase consisting only of the two major subunits was reported[43]. Two oxygen access channels were identified from the structure due to the lack of small subunits, one channel leading to heme *d*, the other to heme $b_{595}$[43]. Unfortunately, the redox potentials of the heme groups were not determined making it impossible to say whether both channels are used. Second, *bd*-II is mainly present as a dimer (Fig. 1) while *bd*-I is only found as a monomer[21,22]. The amino acid residues that mediate the dimer contact in *bd*-II are not fully conserved in *bd*-I

(Supplementary Fig. 6), which may indicate that *bd*-II is a dimer in vivo. During preparation, it turned out that *bd*-II, which is a monomer in LMNG (Supplementary Fig. 1), is much less stable than *bd*-I in detergent. *Bd*-II tends to disintegrate at concentrations below 1 mg/mL and may require higher concentrations to obtain stability by dimerization. Third, the oxygen access channel has a smaller diameter in *bd*-II compared to *bd*-I (Fig. 4). Narrowing the channel in *bd*-I unexpectedly led to a slightly higher specific activity. Thus, a smaller channel diameter could be related to a higher substrate specificity preventing angled molecules to enter deep into the oxygen access channel. The very short proton pathway indicates that substrate protonation at heme *d* in *bd*-II is not rate limiting. Fourth, we have resolved the entire

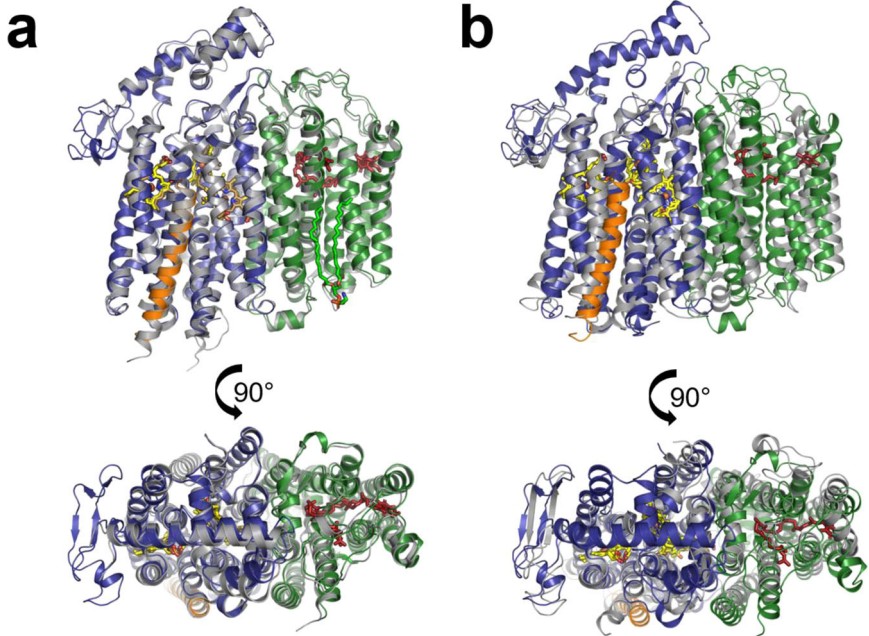

**Fig. 5 Superposition of *bd*-II with *bd*-I and *G. thermodenitrificans bd* oxidase. a** Superimposed structure of *E. coli bd*-II and *E. coli bd*-I and **b** of *E. coli bd*-II and *G. thermodenitrificans bd* oxidase. Subunit AppC is shown in blue, AppB in green and AppX in orange. The homologous subunits of *E. coli bd*-I and *G. thermodenitrificans bd* oxidase are all shown in gray. The heme groups are shown in yellow and ubiquinone-8 in red.

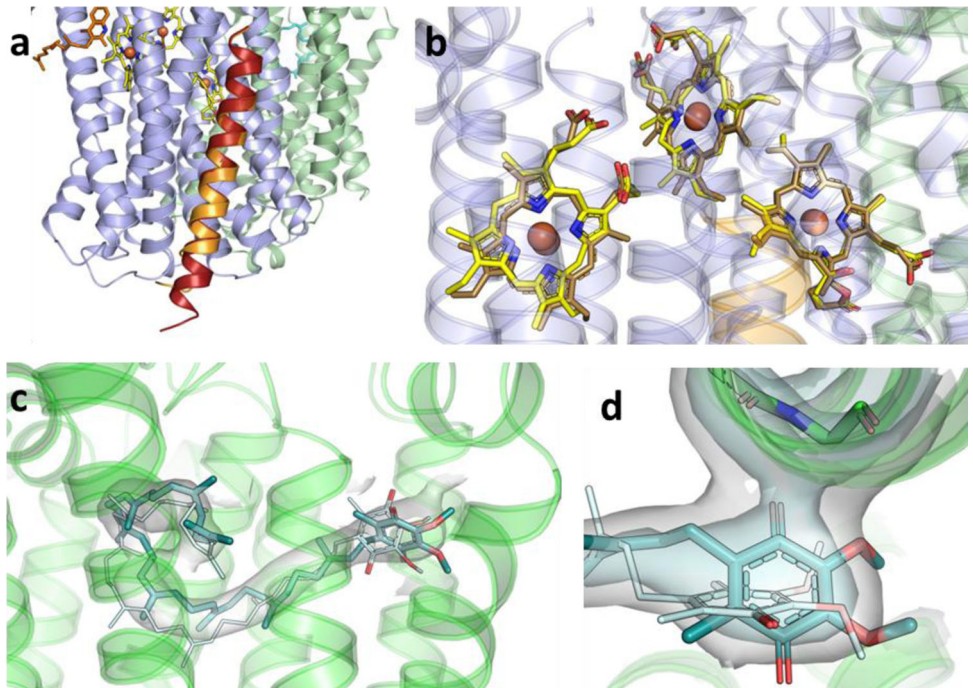

**Fig. 6 Structural similarities between *E. coli bd*-I and *bd*-II. a** Superposition of the small subunits AppX (orange) and CydX (red). **b** Superposition of the heme groups. **c** Superposition of ubiquinone-8 in *E. coli bd*-II (gray, oxygen in red) and *bd*-I (blue, oxygen in red). The Coulomb density map is shown as surface at $1.0\sigma$ (carved at a radius of 2.4 Å around model atom centers). **d** Close-up view of the positioning of the headgroup of ubiquinone-8 in *bd*-I (pale cyan lines) and in *bd*-II (teal sticks). *bd*-II subunit AppB is shown as cartoon. The Coulomb density is given for ubiquinone-8 (bound to AppB) and for neighboring amino acids Ala218[B] and Gly219[B] of AppB at $\sigma$-levels of 1.2 (pale cyan) and 2.0 (light gray), both carved at 2.3 Å radius around model atom centers.

Q-loop of *bd*-II due to binding of aurachin D (Fig. 3). It was speculated that the N-terminal region of the Q-loop is intrinsically disordered or flexible and therefore cannot be resolved in structural analysis[43]. However, as shown here, this is not the case but it should be kept in mind that the incompletely resolved

Q-loop belongs to a *bd* oxidase from the S-subfamily and shows in part a conformation different from that of other *bd* oxidases[43]. Due to its different length the Q-loop can be exchanged between *E. coli bd*-I and *bd*-II but not between *E. coli bd*-I and the *G. thermodenitrificans* enzyme[32,33]. The inhibitor aurachin is bound

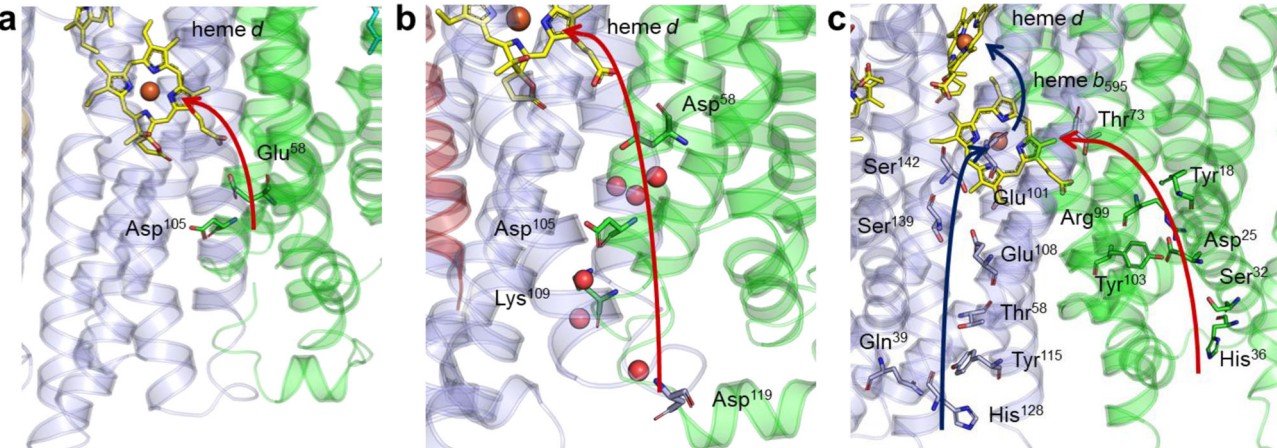

**Fig. 7 Proposed proton pathways to heme *d*. a** shows the short channel in *bd*-II that begins at the end of the large, hydrophilic cavity, **b** the longer proton pathway along a series of titratable amino acid residues in *bd*-I and **c** the two proposed extended proton pathway in *G. thermodenitrificans bd* oxidase[25]. Putative proton pathways are indicated by red arrows that connect titratable residues. The proposed second pathway in *G. thermodenitrificans bd* oxidase is indicated by a blue arrow.

in a hydrophobic pocket provided by the Q-loop and TM helices 6 and 7 of AppC (Fig. 3). The inhibitor and the protein pocket form a very nice complementary structure and binding through several hydrophobic interactions is intensified by a strong H-bond to D239[AppC] (Fig. 3). This position is important for quinol and aurachin binding as the D239N[AppC] variant shows a decreased inhibitor sensitivity (Supplementary Fig. 8) and a clearly diminished duroquinol:oxygen oxidoreductase activity. The influence of this mutation on enzyme activity was reported for *bd*-I[44].

The known structures of bacterial *bd* oxidases revealed a common architecture of *E. coli bd*-I, *bd*-II, *G. thermodenitrificans* and *M. smegmatis bd* oxidase. However, the enzymes seem to be involved in different physiological processes as their genes are expressed at different environmental conditions. The diverging reactivity is exerted by the addition of further, single helix sub-units, by a varying arrangement of the same cofactors or by a different access to the same cofactor. Thus, diverse activities are generated by variations on a common theme. Structure determination of *bd* oxidases from other organisms will expand knowledge about this protein family and give rise to the assignment of new functions of these enzymes that may find their reflection in even more variations of the common *bd* oxidase theme.

## Methods

**Mutagenesis**. KOD polymerase (Merck Millipore, Darmstadt, Germany) was used according to the manufacturer's specifications. Plasmid pET28b(+) was amplified with oligonucleotides pET28b(+)_fwd and pET28b(+)_rev (Supplementary Table 2). The *app*-genes were amplified from genomic DNA from *E. coli* strain BW25113[45] by standard PCR techniques, using the primers listed in Supplementary Table 2. The C-terminus of AppC was decorated with a His-tag affinity peptide. The oligonucleotides contain homologous regions to *appC* or *appX*. Initially, the *appCBX* genes were amplified from *E. coli* strain BW25113 with oligonucleotides appC_fwd and appX_rev containing homologous regions to pET28b(+) (Supplementary Table 2). The resulting PCR products were treated with DpnI and purified by agarose gel electrophoresis. For subsequent AQUA cloning[46], plasmid and insert were mixed in a molar ratio of 1:5. The obtained vector pET28b(+) *appCBX* was used to introduce a sequence encoding a hexa-histidine tag. Next, oligonucleotides appChis_fwd and appB_rev were used to generate the linear fragment *appChisBX* (Supplementary Table 2). This fragment was in turn used as megaprimer for RF cloning[47]. The plasmid encoding *cydAhisBX*[22] was mutated by using the QuikChange protocol to generate the point mutation G100A[bd-I]. To generate *bd*-II variants, we cloned the *appCBX* genes from pET28b(+) into a pBAD vector[31] and transformed *E. coli* strain CBO with the pBAD *appCBX* expression plasmid[20,32]. The pBAD plasmid was mutated by using the QuikChange protocol to generate the point mutation D238N[AppC]. The sequences of the mutagenic primers are listed in Supplementary Table 2. Newly

generated vectors were checked by sequencing (GATC Biotech, Konstanz, Germany). Restriction enzymes were obtained from Thermo Fisher Scientific (Darmstadt, Germany) or Merck Millipore (Darmstadt, Germany).

**Cell growth**. *E. coli* strain BL21*Δcyo[28], lacking the *cyoABCD* genes that encode the ubiquinol: cytochrome *bo₃* oxidase was used as host strain for protein production because the *bo₃* oxidase has a natural His-tag that might interfere with purification of *bd*-II[48]. BL21*Δcyo was transformed with pET28b(+) *appChisBX* by electroporation. Cultures were grown aerobically in 2 L baffled flasks with 800 mL LB-medium each at 37 °C. Gene expression was induced at an $OD_{600}$ of approximately 2 by the addition of 400 μM isopropyl β-D-1-thiogalacto-pyranoside. Cells were harvested 2 h after induction and stored at −80 °C. To produce the *bd*-II D239N[AppC] variant, strain CBO transformed either with pBAD *appCBX* or with pBAD *appCBX*/D239N[AppC] and grown in LB medium as described[20,32]. Cells were harvested 2 h after induction and stored at −80 °C.

**Purification of the *E. coli bd*-II oxidase**. All steps were carried out at 4 °C. 30–40 g shock frozen cells (wet weight) were homogenized in a sixfold volume of 20 mM 3-(N-morpholino)propanesulfonic acid (MOPS), 20 mM NaCl, 0.5 mM phenylmethylsulfonyl fluoride (PMSF), pH 7.0 and a few grains of DNase I using a teflon-in-glass homogenizer. Cells were disrupted with three cycles in a high-pressure homogenizer (1250 bar, Maximator HPL6, Maximator GmbH). After centrifugation (4 °C, 10,000 rpm, 30,000 × g; 20 min, Rotor 25.50, Avanti J-26S XP), membranes were isolated from the supernatant by ultra-centrifugation (4 °C, 50,000 rpm, 250,000 × g, 70 min, Rotor 60Ti, Optima LE80-K, Beckman-Coulter). Subsequently, the sedimented membranes were resuspended in a 1:1 ratio (w/v) with 20 mM MOPS and 20 mM NaCl, pH 7.0 using a teflon-in-glass homogenizer. Finally, the membranes were shock frozen in liquid nitrogen and stored at −80 °C.

6–10 g membranes (wet weight) were thawed and diluted in a sixfold volume 20 mM MOPS, 20 mM NaCl and 0.5 mM PMSF. Membrane proteins were solubilized by incubating the membrane suspension with 1% (w/v) lauryl maltose neopentyl glycol (LMNG) for 90 min and gently stirred at 4 °C. After centrifugation (4 °C, 50,000 rpm, 250,000 × g, 15 min, Rotor 60Ti, Optima LE80-K, Beckman-Coulter), the supernatant was immediately subjected to chromatographic purification. All chromatographic steps were carried out at 4 °C using a NGC Quest 10 Plus (BioRad) chromatography system. 50 mM MOPS, 500 mM NaCl, 500 mM imidazole, 0.5 mM PMSF, 0.003% LMNG, pH 7.0 were added to the supernatant to a final imidazole concentration of 68 mM. This sample was loaded onto a nickel affinity chromatography column (ProBond, 20 mL, Life Technologies) equilibrated in 50 mM MOPS, 500 mM NaCl, 50 mM imidazole, 0.5 mM PMSF, 0.003% LMNG, pH 7.0, at a flow rate of 3 mL/min. The column was washed with the same buffer containing 140 mM imidazole until the absorbance dropped to the baseline. Bound proteins were eluted with 260 mM imidazole. Peak fractions containing *bd*-II were pooled and concentrated to 0.5 mL via ultrafiltration (4 °C, 3800 rpm, 3000 × g, 30 min, Rotor A-4-44, Eppendorf Centrifuge 5804 R, Amico Ultra, 100,000 MWCO, Millipore). The concentrated protein solution was centrifuged (4 °C, 14,000 rpm, 20,800 × g, Eppendorf Centrifuge 5417R) and applied to size exclusion chromatography, using a HiLoad 16/60 Superdex 200 pg column (GE Healthcare), equilibrated in 20 mM MOPS, 20 mM NaCl, 0.5 mM PMSF, 0.003% LMNG, pH 7.0 at a flow rate of 0.3 mL/min. Fractions containing the *bd*-II were pooled and concentrated to 24 mg/mL. Finally, the protein was shock frozen in liquid nitrogen and stored at −80 °C. The *bd*-II oxidase wild-type and the D239N[AppC] variant were prepared according to the same protocol as the proteins produced in BL21* Δcyo. Proteins from strain CBO eluted at identical position from the columns, showed an

identical redox difference spectrum and the same protein band pattern after SDS-PAGE as the proteins produced in strain BL21*Δ*cyo*. The G100A$^{bd-I}$ variant was prepared according to ref. [22].

**Amphipol exchange**. For exchanging the detergent LMNG with amphipol, *bd*-II was supplemented with a threefold mass excess of amphipol A8-35 (Anatrace, Maumee, OH, USA). The sample was incubated at 4 °C for 2 h while stirring. Excess detergent was removed by an addition of a 40-fold mass excess BioBeads (SM-2 resin, BioRad, Feldkirchen, Germany) and further incubation of 3 h at 4 °C under stirring. Excess amphipols were removed by size exclusion chromatography (Superose 6 Increase 10/300 GL, GE Healthcare), equilibrated in 20 mM MOPS, 20 mM NaCl, pH 7.0. Peak fractions were pooled and used for further analysis. For cryo-EM the samples were supplemented with 1 µM aurachin D prior to placing them onto the cryo grids.

**Cryo-electron microscopy and data analysis**. Glow-discharged cryo grids were prepared with 2.5 µL protein solution (5.6 mg/mL) on R1.2/R1.3 Quantifoil holey gold grids. Grids were incubated for 45 s and blotted for 6.5 s prior to plunging into liquid ethane and flash-frozen using a Vitrobot (FEI Company). Cryo-EM recordings were made at the EM Facility at the Rudolf Virchow Center, Würzburg, Germany, with a Titan Krios G3 (Thermo Fisher Scientific) equipped with a Falcon III direct detector. Micrographs were taken at 300 kV with a ×75,000 magnification and a calibrated pixel size of 1.0635 Å. Movies with 47 frames each were collected automatically with EPU at an exposure time of 74.99 s and a total electron exposure of 79 e$^−$/Å$^2$. The final data set contained 1836 movies with a targeted under-focus between 1.4 and 2.4 µm (Supplementary Table 1). All movies were dose weighted and motion-corrected with Motioncorr[49], CTFs were determined using CTFFIND-4.1 [50]. Convolutional neuronal network-based particle picking of all micrographs was conducted with crYOLO[51] using the provided general network. All following steps (Supplementary Fig. 3) were conducted with Relion 3.1.0 [52]. The picked particle positions were imported in Relion and particle images were extracted, followed by a 2D classification. The resulting 2D classes served as template for auto-picking, which resulted in ~800,000 picked particles. The particles were further 2D classified in three rounds. In the first two rounds the particles were arbitrarily separated in six particle subsets and in the third round the particles were separated in three particle subsets. After each round, classes that did neither contain contaminations or empty classes were further processed. These 2D classifications yielded a final particle set of ~100,000 particles. With these particles an initial model was obtained with the InitialModel option in Relion (initial resolution 10 Å and final target resolution 5 Å to ensure the occurrence of helix densities; initial mini-batch size 1000 and final mini-batch size 10,000 for better signal-to-noise). The remaining particles were further refined and purged from residual artifact particles with a 3D classification on the initial model to a subset of ~40,000 particles. The respective 3D class was taken as model for a 3D auto-refinement (in several rounds, later with an imposed C2 symmetry). The remaining ~40,000 particles were 2D classified and the resulting class averages were used as template for another round of particle picking. These new particles were again 2D and 3D classified, and the resulting 120,000 particles were used for a final 3D auto-refinement with following post-processing imposing C2 symmetry to a resolution of 3 Å. The cryo-EM map was taken for protein modeling in COOT 0.8.9.1 [53]. The homologous *bd*-I structure (PDB: 6RX4) was fitted into the map as starting model and mutated to the respective sequence. The model was adjusted to the map and refined with Phenix realspace.refine[52], followed by a validation by MolProbity 4.4 [54]. Model figures were produced with PyMOL 2.4.1.

**UV-vis spectroscopy**. UV-vis difference spectra were recorded with a diode array spectrometer (TIDAS II, J&M Analytik AG). 200 µg protein was diluted in buffer (20 mM MOPS, 20 mM NaCl, 0.003% LMNG, pH 7.0) and a reference spectrum was recorded. The same solution was subsequently reduced with a few grains of dithionite and a UV-vis spectrum of the reduced sample was recorded. The redox spectrum was obtained by subtracting the spectrum of the oxidized sample from that of the reduced one. To determine the binding of CN$^−$, a reference spectrum was recorded, followed by the addition of 0.5 mM KCN to the sample, while spectra were continuously recorded every 30 s. After 30 min, the cyano-complex was reduced by an 80-fold molar excess (0.5 mM) dithionite and the spectrum of the reduced cyano-complex was recorded.

**UV-vis differential titration**. The UV-vis oxidative and reductive titrations were carried out in a thin-layer electrochemical cell as described[23,29] with a dual-beam VARIAN Cary 300 spectrometer coupled to a potentiostat. A gold grid modified with a solution of positively and negatively charged thiols (1:1 solution of cysteamine and mercaptopropionic acid) was used as working electrode, a platinum contact and an aqueous Ag/AgCl 3 M KCl served as counter and reference electrodes, respectively. The *E. coli bd*-II was concentrated in 100 mM KPi, 50 mM NaCl, 0.05% DDM, pH 7.0 to 0.8 mM and 1 h prior to the experiments, the solution was incubated with a mixture of 19 mediators with a final concentration of 25 µM (Supplementary Table 3). The protein sample was deposited on the gold grid and the cell was closed with CaF2 windows and filled with the buffer. All redox difference spectra were recorded at 13 °C in the controlled-potential range of +500/ −500 mV vs. Ag/AgCl. The equilibration time for each potential step was at least 30 min and multiple spectra were recorded to ensure the stabilization of the redox state. The redox potentials of hemes were determined from the modified Nernst equation applied to the experimental titration curves presented as a plot of the absorbance at 562 and 629 nm, each, versus the applied potential.

**Inhibition of *bd*-II oxidase by aurachin C and D**. The duroquinol:oxygen oxidoreductase activity of isolated *E. coli bd*-II oxidase and its inhibition by aurachins C and D were determined by monitoring the O$_2$ consumption with a Clark-type oxygen electrode (Oxyview System, Hansatech Instruments). All steps were conducted at 30 °C. The reaction chamber was filled with 2 mL buffer (20 mM MOPS, 20 mM NaCl, 0.003% LMNG, pH 7.0), 10 µL DTT (1 M) and 5 µL duroquinone in ethanol (100 mM) and incubated for 1 min. 10 µL *bd*-II in LMNG buffer (2 mg/mL) were added to start the reaction. After 2 min, the reaction was inhibited by adding various amounts (1–300 nM) of aurachin C and aurachin D, respectively. The enzymatic activity was corrected for the non-enzymatic activity that was about 5% of the entire activity. Each data point was measured in triplicates. The standard deviation of each measurement is provided.

**Dynamic substrate channel analysis**. As starting point for substrate channel analysis, a conformational ensemble was obtained for both *E. coli bd*-I and *bd*-II through molecular dynamics (MD) simulations with the programs of the Schrödinger software Suite 2021-1 (Schrödinger LLC, NY). First, the cryo-EM structures of *bd*-I (PDB ID: 6RX4) and *bd*-II were prepared with PrepWizard[55] by adding missing hydrogen atoms, adjusting bond orders and formal charges of small-molecule ligands. PROPKA[56] was used to generate the ionization states of polar amino acids at pH 7.0. The structures were energy-minimized using the OPLS4 force field[57]. MD simulations were carried out with Desmond[58]. The optimized structures were embedded in POPC membranes within orthorhombic periodic boundary systems and solvated with TIP3P water molecules, while neutralizing the charge of the systems with counter-ions. After an equilibration protocol, the production of MD simulations was conducted for 100 ns in an NPT ensemble at 300 K regulated by a Nosé–Hoover thermostat and a Martyna–Tobias–Klein barostat. Atomic coordinates were recorded at an interval of 400 ps, for a total 250 frames for each trajectory. Substrate pathways to the active site were analyzed with Caver 3.0 [42] via Caver Analyst 2.0 [59] graphical user-interface. First, the MD frames were stripped of the counter ions, POPC membranes and TIP3P water molecules and used as input for channel computation. During the computation, all protein and ligand atoms were treated as hard spheres. The default threshold value of 0.9 Å was used as bottleneck radius to identify the most probable channels. For the identified tunnels, the residues lining them were equally listed.

**Reporting summary**. Further information on research design is available in the Nature Research Reporting Summary linked to this article.

## Data availability
The cryo-EM maps have been deposited in the Electron Microscopy Data Bank (EMDB; https://www.ebi.ac.uk/emdb) under the accession number EMD-13048. The atomic model has been deposited in the worldwide Protein Data Bank (wwPDB; https://www.wwpdb.org/) under accession number 7OSE. For structural alignments, the PDB entries 5DOQ, 6RKO and 6RX4 were used. Source data are provided with this paper.

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

## Acknowledgements
Electron microscopic data were acquired at the cryo-EM facility of the University Würzburg (DFG equipment grant INST 93/903-1 FUGG to B.B.). This work was supported by the Deutsche Forschungsgemeinschaft by grant -278002225/RTG 2202 to T.F.

## Author contributions
A.G. and J.K. prepared the enzymes and performed the amphipol exchange, J.K. performed enzyme kinetic measurements and made the *bd*-I mutant, T.R. and A.G. made the EM grids, A.G., T.R. and B.B. measured the samples, solved the structure and refined the model, S.O. generated the expression plasmid, A.G., J.K., I.M. and F.M. recorded UV-vis spectra,

R.M. provided the aurachins, I.M., F.M. and P.H. performed the electrochemical titrations, A.F.A.M. and S.G. performed molecular simulations, A.G., J.K. and D.W. made the figures, T.F. designed the project and wrote the paper with corrections from all co-authors.

## Funding

## Competing interests

The authors declare no competing interests.
