## [Peer Review File · Nature Communications]

REVIEWER COMMENTS

Reviewer #1 (Remarks to the Author):

General comments

The manuscript reports a new structure for the bd-II respiratory oxidase from *E. coli*, including the first resolved long Q-loop for this class of respiratory complex. The structural work is novel in parts and displays interesting differences between previously-published structures of bd-type oxidases.

However, there are some shortcomings in the presentation and narrative and there are concerns about how some of the biochemical analyses were performed.

Recurrent amendments:

Italics for heme b and heme d needed throughout

Abstract

Since the structure of bd-I exists, it would be helpful to further emphasise the novelty of this structure.

Introduction

P3 – clarify that the stressors are in references 6-8

P4 – re-phrase ‘that else would be exposed to the Membrane’

P4 – ‘knowledge about bd-II is only scarce’ is a slight overstatement

P4 – Change ‘appBCX genes’ to ‘appCBX genes’

The final paragraph repeats some of the abstract but also covers some additional major findings that would fit better in the abstract.

Results

P5 – re-phrase ‘at aerobic conditions’

P6 – provide extinction coefficients to back up the statement ‘hemes b558, b595 and d in a 1:1:1 stoichiometry’

P6 – Fig. 1 does not actually show the titration data. The data in supplementary Fig. 3 (labelled as Fig 3 incorrectly) show the haem d titration, where the reductive and oxidative titrations do not overlay – these should overlay so I wonder if the discrepancy relates to the binding of oxygen? The peak at 630nm is due to the ferrous deoxygenated form, whereas the trough at 657 nm is due to the oxyferrous form. Can the authors exclude the possibility that these are measuring the transitions of the haem d from oxyferrous to ferrous and then to ferric? Also, in the methods it states that the measured signal is the differential between 629 nm and 562 nm. These reflect redox states of different haem compounds so it is perhaps unsurprising that the transitions do not overlay. Finally, there are no titrations shown that specifically relate to b558 and b595.

P7 – The authors indicate that the gel filtration data in Supplemental figure 1 show a monomer, but there is no calibration scale with standards. This needs including.

P9 - leucin zipper

P14/15 – the authors refer to ‘substrate access to heme d’ and the ‘substrate channel’. They are presumably referring to oxygen access rather than ubiquinol access? Be more specific throughout when referring to oxygen and quinol.

P15 - $\mu\text{mol}/\text{min}\cdot\text{mg}$. This should be per mg (i.e. $/\text{mg}$). Need to provide references for bd-I activity and quote the published specific activity.

Discussion

This is very brief. Need to discuss the data in the context of the literature.

Methods

P24 – need to list mediators or reference exact method.

P25 – the authors add 0.5 mM duroquinone as an electron donor, which will need to become reduced to duroquinol by the 5 mM DTT before it can be an electron donor. However, this combination can produce quite high background rates of oxygen consumption (i.e. in the absence of bd) and it is therefore better to isolate reduced duroquinol for use as an electron donor. Can the authors demonstrate that the background rates were negligible compared to the bd-I-catalysed oxygen consumption rate?

Supplemental Data

Supplemental Figure 2 is labelled as Figure 3

Could make better use of colour in the sequence alignments to aid clarity

Supplemental Figure 7 – to calculate IC50 values it is customary to plot the concentration on a log scale and fit to a sigmoidal function with the magnitude change approximating 100 % activity. This is helpful to the reader and aids visualisation of the transition point. The authors should consider plotting the data in this way, although sub-inhibitory concentrations are required which do not appear to be included.

Reviewer #2 (Remarks to the Author):

Cytochrome bd quinol:O₂ oxidoreductases are respiratory terminal oxidases endowing pathogenic bacteria resistance to cellular stressors thus attractive drug targets. Grauel et. al report the cryo-EM structure of the Escherichia coli cytochrome bd-II type oxidase with the bound inhibitor aurachin D at 3 Å resolution. Although the complex shows an architecture similar to that of bd-I which structures are known, the fully resolved structure of the Q-loop, providing hints for the substrate binding and maybe also the specific binding scheme of inhibitors such as aurachin D, is a significant advance in the field of cytochrome bd research. I recommend this manuscript be published in Nature Communications. Considering the structure description and illustration are quite rough, I have some concerns which should be addressed in revision, as listed below:

1/ Purified bd-II eluted as a “monomer” from size-exclusion chromatography. After amphipol exchange and the addition of inhibitors, predominantly dimeric species of bd-II were detected on the cryo-grid. This is unusual and should be further clarified to exclude the possibility of the artificial oligomerization induced by LMNG or amphipol. LMNG frequently introduces nonspecific oligomers of membrane proteins. Blue Native Page may be a good way to monitor or confirm the population ratio of dimer forms in the sample during purification. Besides, in Supplementary Figure 4, it is not a good way to show all the classes stacked on top of each other. It may be a good choice to lay the main classes side by side to demonstrate whether there are other oligomeric forms as the ones shown in Supplementary Figure 3.

2/ In Supplementary Figure 3, Ab initio 3D reconstruction rarely achieve a high resolution of 5 Å. Meanwhile, the initial model is quite crucial for this kind of unusual dimerization, so maybe you can provide more detailed info on the initial model generation and classification.

3/ The fully resolved structure of the Q-loop is a major highlight in this work. In Figures 3 a and b, the quality of the map and model of the Q-loop should be further clarified with a Supplementary “Model in Map” Figure (to show the map quality for the sidechains).

4/ In Figures 3c and d, maybe you can choose a better or multi-threshold for the map to demonstrate the map/structure quality of aurachin D. Under the current threshold, it is difficult to determine whether the broken discontinuous map signal is aurachin D or to confirm its conformation posture, especially considering that most of the interaction distances are farther than 3.5 Å apart. The only “H-bond of the aurachin nitrogen atom to Asp239 of AppC adds to the strong binding”, is there a way to experimentally demonstrate the contribution of this Asp239 to the binding?

5/ The overall structures of bd-I and bd-II are quite similar, but the orientation of the ubiquinone-8 headgroup are almost perpendicular to each other (in Figure 6c). Could you provide a “model in map” figure for this flexible molecule to justify the rationality of this modelling conformation?

6/ There are some writing problems in the manuscript, such as the consistent use of italics for cytochromes such as b or d and the quotation marks used in the legend of Supplementary Figure 5 which may be a typo.

Point-by-point response to reviewers' comments

Reviewer #1 (Remarks to the Author):

General comments

The manuscript reports a new structure for the bd-II respiratory oxidase from *E. coli*, including the first resolved long Q-loop for this class of respiratory complex. The structural work is novel in parts and displays interesting differences between previously-published structures of bd-type oxidases.

A: We thank the reviewer for these very supportive comments.

However, there are some shortcomings in the presentation and narrative and there are concerns about how some of the biochemical analyses were performed.

Recurrent amendments:

Italics for heme b and heme d needed throughout

A: The names of the individual hemes are now shown in italics throughout the text.

Abstract

Since the structure of bd-I exists, it would be helpful to further emphasise the novelty of this structure.

A: So far, the abstract contained information about the structure and its resolution, the number of subunits, the homologous arrangement of the heme groups and the structural quinone compared to bd-I and that the Q-loop is fully resolved due to the binding of aurachin D. These are all novelties because a structure of bd-II is not available so far. Furthermore, the fully resolved Q-loop and the binding position of an aurachin have not been described before in any bd-oxidase. We now also included a sentence that the fourth subunit found in bd-I is not present in bd-II and that the redox potential of heme d is more positive than in the other bd-oxidases. We kept in mind that the abstract may not extend 150 words.

Introduction

P3 – clarify that the stressors are in references 6-8

A: Several pathogenic bacteria can survive intracellular growth conditions only with a functional bd-oxidase. However, not all cellular stressors causing the need for the presence of a bd-oxidase are known. Our references 6-8 describe the cases when pathogens lacking bd-oxidases cannot survive under distinct growth conditions. The best evidence in the literature is provided for NO and hypoxic conditions as cellular stressors. These are now mentioned in the introduction.

P4 – re-phrase 'that else would be exposed to the Membrane'

A: The sentence has been re-phrased to 'probably plays a functional role by blocking substrate access to heme b_{595} from the membrane'.

P4 – 'knowledge about bd-II is only scarce' is a slight overstatement

A: *This has been replaced by 'much less is known about bd-II'.*

P4 – Change 'appBCX genes' to 'appCBX genes'

The final paragraph repeats some of the abstract but also covers some additional major findings that would fit better in the abstract.

A: *The gene order has been corrected to 'appCBX'. We thank the reviewer for being that careful. We agree with the reviewer that the last paragraph of the introduction summarizes the major new findings of our study, which was intentional. Most of this information is also found in the abstract with the exception of the narrow substrate channel leading to heme d. However, due to the limited space it was not possible to also include this piece of information in the abstract.*

Results

P5 – re-phrase 'at aerobic conditions'

A: *The term was replaced by 'under oxic conditions'*

P6 – provide extinction coefficients to back up the statement 'hemes b558, b595 and d in a 1:1:1 stoichiometry'

A: *The calculation was based on the extinction coefficients provided in our reference 2 that is now cited in the text as 'The (reduced-minus-oxidized) difference spectrum of the preparation showed the typical absorbance of hemes b_{558} , b_{595} and d in a 1:1:1 stoichiometry using the given extinction coefficients².'*

P6 – Fig. 1 does not actually show the titration data. The data in supplementary Fig. 3 (labelled as Fig 3 incorrectly) show the haem d titration, where the reductive and oxidative titrations do not overlay – these should overlay so I wonder if the discrepancy relates to the binding of oxygen? The peak at 630nm is due to the ferrous deoxygenated form, whereas the trough at 657 nm is due to the oxyferrous form. Can the authors exclude the possibility that these are measuring the transitions of the haem d from oxyferrous to ferrous and then to ferric? Also, in the methods it states that the measured signal is the differential between 629 nm and 562 nm. These reflect redox states of different haem compounds so it is perhaps unsurprising that the transitions do not overlay. Finally, there are no titrations shown that specifically relate to b558 and b595.

A: *The redox potentials derived from the electrochemical titrations are shown in Fig. 1 as stated in the text. We now refer two times to Fig. 1 to make clear that it shows the redox potentials of the heme groups and to Supplementary Figure 2 for the titration curves (the figure is now correctly labelled). The description in the 'Methods' might have been expressed somewhat inaccurately: The individual absorbance at 629 nm and 562 nm are plotted against the applied potential. This is now stated more clearly as 'The redox potentials of hemes were determined from the modified Nernst equation applied to the experimental titration curves presented as a plot of the absorbance at 562 and 629 nm, each, versus the applied potential.' Furthermore, we have included additional titration curves including the titration that relates to the b-type heme groups with more data points in order to complete data presentation (oxidative and reductive titration). Data are measured from the fully oxidized to the fully reduced state of the bd oxidase (and vice versa). The titration curves are recorded in a closed thin layer electrochemical cell under anoxic condition, thus, there is no oxygen available for the d heme and any possible reaction intermediate. Due to the anoxic conditions, it can be excluded that the desoxygenated d heme becomes oxygenated during the titration and because the*

oxidative and reductive titrations result in the same redox potential it is evident that the titrations do not include transitions of the heme d from oxyferrous to ferrous and then to the ferric form. Please note that the experiments take very long (about one day), since the equilibration for each titration point takes about 30-40 minutes. This is often seen in cytochrome oxidases. But most importantly, the curves have been reproduced several times.

P7 – The authors indicate that the gel filtration data in Supplemental figure 1 show a monomer, but there is no calibration scale with standards. This needs including.

A: A calibration curve is now shown in Supplementary Figure 1 as Figure 1b (together with a BN-PAGE, see below). Both techniques clearly demonstrate that the preparation in LMNG only contains the monomeric form of bd-II oxidase.

P9 - leucin zipper

A: Has been corrected to 'leucine zipper'.

P14/15 – the authors refer to 'substrate access to heme d' and the 'substrate channel'. They are presumably referring to oxygen access rather than ubiquinol access? Be more specific throughout when referring to oxygen and quinol.

A: Yes, when talking about the heme d we refer to oxygen as substrate. To be more precise, the word 'substrate' in the context of 'channel' or 'access' has been replaced by 'oxygen' in all cases on p 14/15.

P15 - $\mu\text{mol}/\text{min} \cdot \text{mg}$. This should be per mg (i.e. /mg). Need to provide references for bd-I activity and quote the published specific activity.

A: We are talking about specific enzyme activity that is given as U/mg protein. With $U = \mu\text{mol substrate}/\text{min}$, we get $\mu\text{mol}/\text{min} \cdot \text{mg}$, for clarity we know write $\mu\text{mol}/(\text{min} \cdot \text{mg})$. We did not find any value in the literature that describes the duroquinol oxidase activity of the isolated E. coli bd-I oxidase. Many measurements were done with the enzyme in the membrane. The values obtained with these measurements cannot be compared to our measurements due to the uncertainty of the enzyme concentration in the membrane. The few measurements that have been undertaken with the isolated enzyme used ubiquinol-1 as substrate, not duroquinol. In most cases v_{max} values are reported. However, we just would like to determine the effect of the mutation on the enzymes function and for this we are simply comparing wild-type and variant enzymes at one given substrate concentration to each other. We would be very grateful if the reviewer could supply us with a reference for the duroquinol oxidase activity of an isolated E. coli bd-I oxidase at the given substrate concentrations.

Discussion

This is very brief. Need to discuss the data in the context of the literature.

*A: We have extended the discussion and especially included a comparison with the just recently published structure of bd oxidase from Mycobacterium smegmatis (Wang, W. et al. Nature Commun. **12**, 4621 (2021).*

Methods

P24 – need to list mediators or reference exact method.

A: We added a list of the mediators, their redox potentials, solvents and sources as Supplementary Table 3.

P25 – the authors add 0.5 mM duroquinone as an electron donor, which will need to become reduced to duroquinol by the 5 mM DTT before it can be an electron donor. However, this combination can produce quite high background rates of oxygen consumption (i.e. in the absence of bd) and it is therefore better to isolate reduced duroquinol for use as an electron donor. Can the authors demonstrate that the background rates were negligible compared to the bd-I-catalysed oxygen consumption rate?

A: The non-enzymatic reaction is less than 5% of the enzymatic reaction as shown below:

The diagram shows a typical kinetic trace with buffer containing 0.25 mM Q_0 and 5 mM DTT as described in the 'Methods' section (a). The non-enzymatic activity is 0.2 $\mu\text{mol}/(\text{min mg})$. After the addition of bd II (b) the activity is 3.86 $\mu\text{mol}/(\text{min mg})$. The non-enzymatic activity is about 5% of the entire activity. The small amount of the non-enzymatic activity was always measured and the value was subtracted from the total activity to obtain the enzymatic activity. This is now mentioned in the 'Methods' section: 'The enzymatic activity was corrected for the non-enzymatic activity that was about 5% of the entire activity.'

Supplemental Data

Supplemental Figure 2 is labelled as Figure 3

A: This has been corrected.

Could make better use of colour in the sequence alignments to aid clarity'

A: Amino acid residues involved in heme, oxygen and aurachin binding, residues of the Q-loop, residues involved in the dimerization of bd-II and residues discussed to be involved in proton translocation are now marked with colours in Supplemental Figure 6.

Supplemental Figure 7 – to calculate IC₅₀ values it is customary to plot the concentration on a log scale and fit to a sigmoidal function with the magnitude change approximating 100 % activity. This is helpful to the reader and aids visualisation of the transition point. The authors should consider plotting the data in this way, although sub-inhibitory concentrations are required which do not appear to be included.

A: We thank the reviewer for that excellent hint. Now, we show the direct plot as well as the log plot in Supplemental Figure 7. According to the different plot, the IC50 values slightly changed from 7.4 to 7.1 nM (aurachin C) and from 6.5 to 11.1 nM (aurachin D). This is now also corrected in the main text (p. 11). In addition, we show both plots of the inhibition of the D239N^{AppC} variant by aurachin D as requested by reviewer 2 (see below).

Reviewer #2 (Remarks to the Author):

Cytochrome bd quinol:O₂ oxidoreductases are respiratory terminal oxidases endowing pathogenic bacteria resistance to cellular stressors thus attractive drug targets. Grauel et. al report the cryo-EM structure of the Escherichia coli cytochrome bd-II type oxidase with the bound inhibitor aurachin D at 3 Å resolution. Although the complex shows an architecture similar to that of bd-I which structures are known, the fully resolved structure of the Q-loop, providing hints for the substrate binding and maybe also the specific binding scheme of inhibitors such as aurachin D, is a significant advance in the field of cytochrome bd research. I recommend this manuscript be published in Nature Communications.

A: We thank the reviewer for these very supportive comments.

Considering the structure description and illustration are quite rough, I have some concerns which should be addressed in revision, as listed below:

1/ Purified bd-II eluted as a “monomer” from size-exclusion chromatography. After amphipol exchange and the addition of inhibitors, predominantly dimeric species of bd-II were detected on the cryo-grid. This is unusual and should be further clarified to exclude the possibility of the artificial oligomerization induced by LMNG or amphipol. LMNG frequently introduces nonspecific oligomers of membrane proteins. Blue Native Page may be a good way to monitor or confirm the population ratio of dimer forms in the sample during purification. Besides, in Supplementary Figure 4, it is not a good way to show all the classes stacked on top of each other. It may be a good choice to lay the main classes side by side to demonstrate whether there are other oligomeric forms as the ones shown in Supplementary Figure 3.

A: The oligomeric state of the preparation in LMNG is now shown in Supplementary Figure 1 as Figure 1b. Here, a calibration curve of the size exclusion column is included as well as a BN-gel of the preparation. Both techniques unequivocally show that the preparation of bd-II oxidase in LMNG is only present in the monomeric form. However, we cannot fully exclude the possibility that the amphipol exchange leads to a dimerisation although we are not aware of any report on this in the literature. In contrast, we have shown with mass photometry that oligomeric state of several membrane proteins is not influenced by exchanging detergent by amphipol (our ref. 31 and Olerinyova et al., Chem. (2021) 7:224-236, doi: 10.1016/j.chempr.2020.11.011 A). We added a sentence on p. 7: ‘Although unlikely³¹, we cannot completely exclude the possibility that the presence of amphipol led to protein dimerization.’ Supplementary Figure 4 has been corrected according to the proposal of the reviewer. The classes are now shown side-by-side. However, this does not help to identify other oligomeric forms of bd-II oxidase because the monomeric particles are already excluded by 2D classification. Generation of the initial model was only possible with the dimeric particles.

2/ In Supplementary Figure 3, Ab initio 3D reconstruction rarely achieve a high resolution of 5 Å. Meanwhile, the initial model is quite crucial for this kind of unusual dimerization, so maybe you can provide more detailed info on the initial model generation and classification.

A: We thank the reviewer for pointing us to this issue. We have clarified this point in the 'methods' section on p. 21/22 and clearly stated that 5 Å were set as target resolution to ensure the occurrence of helix densities. It is now written: 'With these particles an initial model was obtained with the InitialModel option in Relion (initial resolution 10 Å and final target resolution 5 Å to ensure the occurrence of helix densities; initial mini-batch size 1'000 and final mini-batch size 10'000 for better signal-to-noise)'.

3/ The fully resolved structure of the Q-loop is a major highlight in this work. In Figures 3 a and b, the quality of the map and model of the Q-loop should be further clarified with a Supplementary "Model in Map" Figure (to show the map quality for the sidechains).

A: We have included a new Supplementary Figure 7, in which we show the 'model-in-map' of a part of the Q-loop to enable the reader to get an impression on the quality of the map.

4/ In Figures 3c and d, maybe you can choose a better or multi-threshold for the map to demonstrate the map/structure quality of aurachin D. Under the current threshold, it is difficult to determine whether the broken discontinuous map signal is aurachin D or to confirm its conformation posture, especially considering that most of the interaction distances are farther than 3.5 Å apart.

A: We have revised Figures 3c and 3d and added maps at 0.5 σ and 1.0 σ to the 0.65 σ map presented in the first version of the manuscript (all carved at 2 Å radius around model atoms). The legend of the figure was revised accordingly.

The only "H-bond of the aurachin nitrogen atom to Asp239 of AppC adds to the strong binding", is there a way to experimentally demonstrate the contribution of this Asp239 to the binding?

A: We tried to support this statement by site-directed mutagenesis. So far, bd-II is expressed from a pET plasmid and produced in strain BL21 Δ cyo. However, we were not able to produce a bd-II variant in strain BL21* Δ cyo as the appC gene copy on the plasmid was always replaced by the chromosomal version of the appC gene by a precise recombination. Therefore, we cloned the app-operon into a pBAD plasmid and produced the variants in strain CBO that lacks the chromosomal version of appCBX. The wild type protein and the variant were produced in that strain and could be isolated by the same chromatographic procedures without any significant differences. The enzymatic activity of the isolated D239N^{AppC} variant and its inhibition by aurachin D was determined and compared to the wild-type protein also expressed from the pBAD plasmid and produced in the CBO strain. It turned out that the duroquinol oxidase activity of the D239N^{AppC} variant was indeed only 23% of that of the wild-type and that IC₅₀ is nearly six times higher (61 instead of 11 nM). These are clear evidences that Asp239 is involved in quinol oxidation and aurachin binding. These data are now shown on p. 10 in the 'Results' section and on pages 18 and 20 in the 'Methods' section.*

5/ The overall structures of bd-I and bd-II are quite similar, but the orientation of the ubiquinone-8 headgroup are almost perpendicular to each other (in Figure 6c). Could you provide a "model in map" figure for this flexible molecule to justify the rationality of this modelling conformation?

A: We changed Figure 6 by adding maps for ubiquinone-8 and neighbouring Ala218^B and Gly219^B at 1.2 and 2.0 σ (all carved at 2.3 Å). Furthermore, we added a new part to the figure, namely Figure 6d that shows a close up of the headgroup with a 'model-in-map' to justify the modelled quinone

conformation. The legend of the figure was changed accordingly.

6/ There are some writing problems in the manuscript, such as the consistent use of italics for cytochromes such as b or d and the quotation marks used in the legend of Supplementary Figure 5 which may be a typo.

A: The names of the cytochromes are now consistently written in italics and the quotation marks are deleted in the legend of Supplementary Figure 5. The text was corrected by a native speaker.

REVIEWERS' COMMENTS

Reviewer #1 (Remarks to the Author):

The authors have done a thorough job of addressing the comments and the manuscript is now suitable for publication. One very minor issue that remains is the expression of specific activity as $\mu\text{mol}/\text{min}\cdot\text{mg}$. I still don't understand why you are multiplying by the mg of protein - surely is 'umol per min per mg' in which case you would divide by the mg of protein (i.e. $\text{umol}/\text{min}/\text{mg}$)?

Reviewer #2 (Remarks to the Author):

Although I still have great doubts about the reliability of the ligand conformation described in Figure 6d, which panel does not seem to be referenced anywhere in the manuscript, the authors did address most of the concerns, especially the great efforts to carry out validation experiments and obtain convincing results. I think this paper can be considered for publication now.

Point-by-point response to reviewers' comments

Reviewer #1 (Remarks to the Author):

The authors have done a thorough job of addressing the comments and the manuscript is now suitable for publication. One very minor issue that remains is the expression of specific activity as $\mu\text{mol}/\text{min}\cdot\text{mg}$. I still don't understand why you are multiplying by the mg of protein - surely it is 'umol per min per mg' in which case you would divide by the mg of protein (i.e. $\mu\text{mol}/\text{min}/\text{mg}$)?

A: The specific activity is now expressed in units of " $\mu\text{mol}/\text{min}/\text{mg}$ ".

Reviewer #2 (Remarks to the Author):

Although I still have great doubts about the reliability of the ligand conformation described in Figure 6d, which panel does not seem to be referenced anywhere in the manuscript, the authors did address most of the concerns, especially the great efforts to carry out validation experiments and obtain convincing results. I think this paper can be considered for publication now.

A: A reference to Figure 6d is now provided in the main text.